# Terminal Distributed Cooperative Guidance Law for Multiple UAVs Based on Consistency Theory

Zhanyuan Jiang [1], Jianquan Ge [1,*], Qiangqiang Xu [2] and Tao Yang [1]

1   College of Aerospace Science and Engineering, National University of Defense Technology, Changsha 410073, China; jiangzhanyuan@nudt.edu.cn (Z.J.); yt_yangtao@nudt.edu.cn (T.Y.)
2   College of Space Command, Space Engineering University, Beijing 101416, China; 2003010218@st.btbu.edu.cn
*   Correspondence: nudt_gejq@nudt.edu.cn; Tel.: +86-15290198937

**Abstract:** In order to realize a saturation attack of multiple unmanned aerial vehicles (UAVs) on the same target, the problem is transformed into one of multiple UAVs hitting the same target simultaneously, and a terminal distributed cooperative guidance law for multiple UAVs based on consistency theory is proposed. First, a new time-to-go estimation method is proposed, which is more accurate than the existing methods when the leading angle is large. Second, a non-singular sliding mode guidance law (NSMG) of impact time control with equivalent control term and switching control term is designed, which still appears to have excellent performance even if the initial leading angle is zero. Then, based on the predicted crack point strategy, the NSMG law is extended to attack maneuvering targets. Finally, adopting hierarchical cooperative guidance architecture, a terminal distributed cooperative guidance law based on consistency theory is designed. Numerical simulation results verify that the terminal distributed cooperative guidance law is not only applicable to different forms of communication topology, but also effective in the case of communication topology switching.

**Keywords:** UAVs; impact time control; sliding mode control; cooperative guidance law; consistency theory





## 1. Introduction

With the rapid iterative update of the air and antimissile defense system equipped around an enemy's high-value targets, it becomes increasingly difficult for a single unmanned aerial vehicle (UAV) to attack high-value targets [1]. In order to solve this problem, there are usually two solutions: one is to adopt a cluster cooperative attack to break through with intelligent cooperation and quantitative advantage; the second is to break through with a speed advantage [2]. For the first, an effective method to achieve multiple UAVs cooperative attack is to control the impact time, which will realize the simultaneous attack of multiple UAVs on targets, thereby improving the impact effect [3].

The design of the impact time control guidance law is actually a tracking problem in which the final impact time error is the tracking error. After defining the impact time error, many system control theories, such as bias proportional guidance with error feedback, sliding mode control theory, Lyapunov function, etc., can be used to make the tracking error zero [4–6]. In [7], a guidance law with impact time control was proposed for the first time in 2006, which consists two parts: one is the classic proportional navigation guidance (PNG), and the other is the feedback item of impact time error.

In [8,9], considering the impact angle constraint, the fast terminal sliding mode algorithm is applied to meet the requirements of guidance accuracy and landing angle by adjusting the line of sight angular velocity. In [10], the second-order sliding mode control theory was used to make the time-to-go estimation curve converge to the desired time-to-go curve in finite time. On this basis, the desired time-to-go was planned by using a double-layer cooperative guidance structure, so as to meet the impact time cooperative guidance of multiple aircraft. In [11], the space is expanded from two-dimensional to

three-dimensional, and a three-dimensional impact time control cooperative guidance law satisfying the line-of-sight constraint was proposed. Refs. [12,13] proposed a guidance law training framework based on reinforcement learning theory, which was robust to uncertainties and different parameters.

The above research mainly focuses on the cooperative guidance laws for stationary targets, and there is relatively little research on the cooperative guidance laws for maneuvering targets. References [14–21] study the problem of cooperative guidance for maneuvering targets, but reference [14] needs to assume that the tracking equation can meet the linearization condition of small disturbance. References [14–19] need to assume that the direct measurement information of target acceleration can be obtained, which is usually difficult to achieve in engineering practice; although reference [20] studies the cooperative guidance of maneuvering targets, its method is centralized. The method adopted in reference [21] requires that the communication topology is undirected, which usually leads to more traffic and energy consumption.

In recent years, when the terminal impact angle constraint has been considered at the same time, the impact time and angle control guidance law has gradually developed. Based on the non-singular terminal sliding mode control theory (NTSMC), a guidance law satisfying both impact time and impact angle constraints was designed in [22]. Compared with the traditional sliding mode guidance law, the proposed guidance law did not need to design the line of sight curve offline, nor did it need to switch between the impact time control guidance law and the impact angle control guidance law. In [23], an impact angle control guidance law was designed based on backstepping control method, and an impact time control guidance law was designed based on proportional guidance. The constraints of impact time and impact angle were finally satisfied by using segments. In [24], a conversion scheme was designed. When the impact time error was greater than a certain specified value, the impact time control guidance law based on sliding mode theory was adopted. When the impact time error was less than a certain fixed value, the optimal guidance law designed in [25] satisfying the impact angle constraint was adopted in order to finally realize the cooperative control of impact time and angle. In [26], the trajectory optimization problem with impact time and impact angle constraints was firstly transformed into a nonlinear trajectory planning problem, and then the Gauss pseudo-spectral method was adopted to solve the problem with the optimization objective of minimizing the total control energy. Reference [27] proposed a two-stage guidance law with auxiliary stage. By appropriately modifying the switching conditions of the two-stage guidance law with auxiliary stage, the impact time and angle can be controlled at the same time.

Considering the mutual communication among aircraft, in [28] the average estimated value of the time-to-go of each member was taken as the coordination variable to design the variation curve of range, and the control quantity was designed to track the nominal trajectory, so as to realize the impact time cooperative guidance. Based on the principle of distributed communication and network synchronization, a distributed time cooperative guidance law was designed by taking the "lead-followers" mode to realize the simultaneous convergence of multiple aircrafts to the target position in [29]. In [30], the desired time-to-go was directly set as the average of each member's time-to-go, so as to design a hybrid guidance law satisfying both impact time and impact angle. The research in [31] designed a guidance and control integrated guidance law satisfying the impact time constraint, in which not only the time-varying velocity, but also the constraints such as uncertainty and line-of-sight were considered.

The above cooperative guidance laws based on communication adopted a centralized coordination strategy, and the coordination variable existed only in one member of the formation, which easy to implement. However, this strategy required the information of the whole formation, and when the members were attacked and failed, the coordinated control of the entire formation would fail, which would reduce the robustness and reliability of the system. Therefore, a distributed cooperative guidance law design based on

consistency theory is proposed in this paper. The main contributions of this paper are as follows:

(1) A new time-to-go estimation method is proposed, which is more accurate than the existing method in [32] when the leading angle is large.

(2) A non-singular sliding mode guidance law (NSMG) of impact time control with equivalent control term and switching control term is designed, which still appears to have excellent performance even if the initial leading angle is zero, while some existing impact time control laws in [4,8,33] are invalid. Then the guidance law is extended to attack maneuvering targets.

(3) Adopting hierarchical cooperative guidance architecture, a terminal distributed cooperative guidance law based on consistency theory is designed, which is not only applicable to different forms of communication topology, but also effective in the case of communication topology switching.

The other parts of this paper are arranged as follows: In Section 2, the problem statement and motion models are given. The new time-to-go estimation method and the bottom layer guidance law based on sliding mode control theory are proposed in Section 3. The upper-level distributed coordination strategy based on the consistency theory is given in Section 4. Several numerical simulation examples are provided and compared in Section 5. The conclusions are given in the final section.

## 2. Problem Statement

Two points are explained before establishing the cooperative guidance model. First, during the flight, the thrust of the UAV is adjusted in a small range, which can keep the velocity of the UAV basically unchanged, and the terminal attack distance is short, usually only a few kilometers to more than 10 kilometers. Therefore, it can be assumed that the velocity of each UAV is a constant. Second, in the process of designing the guidance law, the guidance law can be designed separately in the longitudinal plane and the horizontal plane. The guidance law designed in the longitudinal plane is to keep the UAV flying at a fixed height, and the cooperative guidance law designed in the horizontal plane to is meet the relevant cooperative strike requirements. Therefore, it can be assumed that the UAV and the target are in the same horizontal plane. Therefore, four assumptions can be made, as below:

**Assumption 1:** *The UAV velocity can be considered as a constant value.*

**Assumption 2:** *The UAV and target are considered as ideal point-mass models.*

**Assumption 3:** *The target is stationary.*

**Assumption 4:** *Compared with the guidance loop dynamics of UAV, the dynamic response speed of the UAV detection device and autopilot is fast enough, so it can be ignored.*

Based on the above assumptions, it is assumed that the UAV in the two-dimensional plane attacks the stationary target at a constant speed, and the relative motion relationship is shown in Figure 1.

The UAV and the target are denoted by $M$ and $T$, respectively. The equations describing the motions between the UAV and the target can be expressed as follows:

$$\dot{r} = -V_M \cos \sigma_M \tag{1}$$

$$\dot{\lambda} = -\frac{V_M \sin \sigma_M}{r} \tag{2}$$

$$\dot{\gamma}_M = a_M / V_M \tag{3}$$

$$\sigma_M = \gamma_M - \lambda \tag{4}$$

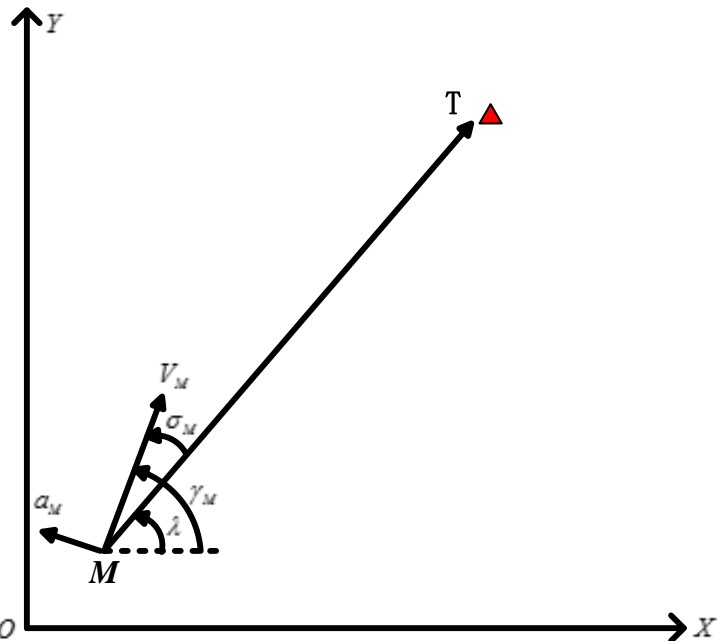

**Figure 1.** Relative motion relationship between unmanned aerial vehicle (UAV) and target.

In the above equations, $V_M$ is the UAV velocity, the symbol $r$ is the relative distance between the UAV and target, namely the range-to-go. Symbols $\gamma_M$, $\lambda$ and $\sigma_M$ represent the flight path angle, the line of sight (LOS) angle and the leading angle, respectively. Symbol $a_M$ is the acceleration command.

## 3. Design of Bottom Guidance Law Based on Sliding Mode Control Theory

In this section, a new time-to-go estimation method is first proposed and compared with the method in [32]. Then, the NSMG law for impact time control based on sliding mode control theory is designed.

### 3.1. Time-to-Go Estimation of PNG Law

Assuming that the UAV is guided by the PNG law, the acceleration is expressed as follows:

$$a_M = NV_M\dot{\lambda} \tag{5}$$

where, $N$ is the navigation gain and $\dot{\lambda}$ is the rate of the LOS angle.

Substituting Equation (5) into Equation (3), yields

$$\dot{\gamma}_M = N\dot{\lambda} \tag{6}$$

Differentiating Equation (4) and substituting Equation (6), yields:

$$\dot{\sigma}_M = \dot{\gamma}_M - \dot{\lambda} = (N-1)\dot{\lambda} \tag{7}$$

Substituting Equation (2) into Equation (7), yields:

$$\dot{\sigma}_M = -\frac{(N-1)V_M\sin\sigma_M}{r} \tag{8}$$

It can be obtained from Equations (1) and (8) that:

$$\frac{d\sigma_M}{dr} = \frac{\dot{\sigma}_M}{\dot{r}} = \frac{(N-1)\tan\sigma_M}{r} \tag{9}$$

Integrating Equation (9) and its solution can be obtained as follows:

$$r = r_0 \left( \frac{\sin \sigma_M}{\sin \sigma_{M0}} \right)^{\frac{1}{N-1}}$$

(10)

where $r_0$ is the initial relative distance and $\sigma_{M0}$ is the initial leading angle.

Substituting Equation (10) into Equation (8), yields

$$\dot{\sigma}_M = -(N-1)V_M \sin \sigma_M / r_0 \left( \frac{\sin \sigma_M}{\sin \sigma_{M0}} \right)^{\frac{1}{N-1}} = K(\sin \sigma_M)^{\frac{N-2}{N-1}}$$

(11)

where $K = -\frac{(N-1)V_M}{r_0}(\sin \sigma_{M0})^{\frac{1}{N-1}}$.

It can be obtained from Equation (11) that:

$$dt = \frac{1}{K}(\sin \sigma_M)^{\frac{2-N}{N-1}} d\sigma_M$$

(12)

Integrating Equation (12) and using Taylor series expansion, ignore advanced items, $\sin x = x - \frac{1}{6}x^3$ and $(1+x)^\alpha = 1 + \alpha x$, yields:

$$
\begin{aligned}
t - t_0 \quad &= \frac{1}{K}\int_{\sigma_{M0}}^{\sigma_M} (\sin \sigma_M)^{\frac{2-N}{N-1}} d\sigma_M \\
&\approx \frac{1}{K}\int_{\sigma_{M0}}^{\sigma_M} \left( \sigma_M - \frac{\sigma_M^3}{6} \right)^{\frac{2-N}{N-1}} d\sigma_M \\
&= \frac{1}{K}\int_{\sigma_{M0}}^{\sigma_M} \sigma_M^{\frac{2-N}{N-1}} \left( 1 - \frac{\sigma_M^2}{6} \right)^{\frac{2-N}{N-1}} d\sigma_M \\
&\approx \frac{1}{K}\int_{\sigma_{M0}}^{\sigma_M} \sigma_M^{\frac{2-N}{N-1}} \left( 1 - \frac{2-N}{N-1}\frac{\sigma_M^2}{6} \right) d\sigma_M \\
&= \frac{1}{K}\int_{\sigma_{M0}}^{\sigma_M} \left( \sigma_M^{\frac{2-N}{N-1}} + \frac{2-N}{N-1}\frac{\sigma_M^{\frac{N}{N-1}}}{6} \right) d\sigma_M
\end{aligned}
$$

(13)

Equation (13) can be further simplified as follows:

$$t = t_0 + \frac{r_0}{V_M}\left( 1 + \frac{2-N}{6(N-1)(2N-1)}\sigma_{M0}^2 \right)\left( \frac{\sigma_{M0}}{\sin \sigma_{M0}} \right)^{\frac{1}{N-1}} - \frac{r_0}{V_M}\left( 1 + \frac{2-N}{6(N-1)(2N-1)}\sigma_M^2 \right)\left( \frac{\sigma_M}{\sin \sigma_{M0}} \right)^{\frac{1}{N-1}}$$

(14)

When the UAV attacks the target, the leading angle $\sigma_M$ equals zero. Therefore, the time-to-go $t_{go}$ at the moment $t$ can be expressed as follows:

$$t_{go} = \frac{r}{V_M}\left( 1 + \frac{2-N}{6(N-1)(2N-1)}\sigma_M^2 \right)\left( \frac{\sigma_M}{\sin \sigma_M} \right)^{\frac{1}{N-1}}$$

(15)

Defining,

$$N\prime = \frac{2-N}{6(N-1)(2N-1)}$$

(16)

and the new time-to-go estimation method proposed in Equation (15) can be rewritten as follows:

$$t_{go} = \frac{r}{V_M}\left( 1 + N\prime\sigma_M^2 \right)\left( \frac{\sigma_M}{\sin \sigma_M} \right)^{\frac{1}{N-1}}$$

(17)

Here, another time-to-go estimation method proposed in [32] is also given as below:

$$t_{go} = \frac{r}{V_M}\left( 1 + \frac{\sigma_M^2}{2(2N-1)} \right)$$

(18)

*3.2. Design of the Impact Time Control Guidance Law*

3.2.1. Design of the Guidance Law for Stationary Target

For stationary targets, considering the impact time control, the sliding mode surface is designed as below:

$$s = t_f - t_f^d = t + t_{go} - t_f^d = t_{go} - t_{go}^d \tag{19}$$

where, $t_f^d$ and $t_{go}^d$ are the desired impact time and the desired time-to-go respectively. $t_{go}$ is the time-to-go under proportional navigation law and the expression is shown in Equation (18).

The time derivative of Equation (19) is expressed as follows:

$$
\begin{aligned}
\dot{s} &= \dot{t}_{go} - \dot{t}_{go}^d \\
&= (1 + K_1) + (K_2 + K_3)\dot{\sigma}_M \\
&= (1 + K_1) + (K_2 + K_3)\left(\frac{a_M}{V_M} + \frac{V_M \sin \sigma_M}{r}\right) \\
&= (1 + K_1) + (K_2 + K_3)\frac{V_M \sin \sigma_M}{r} + (K_2 + K_3)\frac{a_M}{V_M}
\end{aligned}
\tag{20}
$$

where, $K_1$, $K_2$ and $K_3$ are the corresponding coefficients, and the specific expressions can be expressed as:

$$K_1 = -\cos \sigma_M \left(1 + N\prime \sigma_M^2\right)\left(\frac{\sigma_M}{\sin \sigma_M}\right)^{\frac{1}{N-1}} \tag{21}$$

$$K_2 = \frac{r}{V_M}\frac{1}{N-1}\left(\frac{\sigma_M}{\sin \sigma_M}\right)^{\frac{1}{N-1}-1}\frac{\sin \sigma_M - \sigma_M \cos \sigma_M}{\sin^2 \sigma_M}\left(1 + N\prime \sigma_M^2\right) \tag{22}$$

$$K_3 = \frac{r}{V_M}(2N\prime \sigma_M)\left(\frac{\sigma_M}{\sin \sigma_M}\right)^{\frac{1}{N-1}} \tag{23}$$

According to the sliding surface designed by Equation (19), the impact time control guidance law based on Lyapunov non-linear control theory is designed as follows:

$$a_M = a_M^{eq} + a_M^{sw} \tag{24}$$

$$a_M^{eq} = -\frac{V_M}{K_2 + K_3}\left((1 + K_1) + (K_2 + K_3)\frac{V_M \sin \sigma_M}{r}\right) \tag{25}$$

$$a_M^{sw} = -ks \sin \sigma_M - M(p\,\mathrm{sign}(K_2 + K_3) + 1)\mathrm{sign}(s) \tag{26}$$

where, $a_M^{eq}$ and $a_M^{sw}$ are the equivalent part and switching part of the guidance law, respectively, and the parameters $k > 0, M > 0, p > 1$. The equivalent control item is used to control the line-of-sight angular velocity to ensure that the UAV can impact the target, and to maintain the sliding mode surface reaching law $\dot{s} = 0$. The switching control term is to satisfy the impact time constraint, while ensuring that the designed sliding mode guidance law Equation (24) satisfies the Lyapunov stability condition as well as being non-singular, that is, not containing singular points.

3.2.2. Proof of Stability

Choose $V = (1/2)s^2$ as the Lyapunov function, then,

$$\dot{V} = s\dot{s} = -\frac{K_2 + K_3}{V_M}ks^2 \sin \sigma_M - \frac{M}{V_M}(p|K_2 + K_3| + (K_2 + K_3))|s| \tag{27}$$

It can be seen from $(K_2 + K_3)\sin \sigma_M \geq 0$ and $p|K_2 + K_3| + (K_2 + K_3) \geq 0$ that $\dot{V}$ is negative semidefinite, which means that when the leading angle, the sliding surface $s = 0$ may not be satisfied. In order to satisfy the attack time control constraints and make the sliding surface $s = 0$, it is necessary to explain that the leading angle $\sigma_M = 0$ is not an attractor.

It can be seen from Equation (4) that:

$$\dot{\sigma}_M = \dot{\gamma}_M - \dot{\lambda} \tag{28}$$

When the leading angle $\sigma_M = 0$, it can be seen from Equation (2) that the rate of change of line of sight angle is as follows:

$$\dot{\lambda} = -\frac{V_M \sin \sigma_M}{r} = 0 \tag{29}$$

Equivalent control term of the guidance law:

$$a_M^{eq} = -\frac{V_M}{K_2 + K_3}\left((1 + K_1) + (K_2 + K_3)\frac{V_M \sin \sigma_M}{r}\right) = 0 \tag{30}$$

Switching term of the guidance law:

$$a_M^{sw} = -ks \sin \sigma_M - M(p\,\mathrm{sign}(K_2 + K_3) + 1)\mathrm{sign}(s) = -M\mathrm{sign}(s) \tag{31}$$

It can be obtained from Equation (3) that:

$$\dot{\gamma}_M = \frac{a_M}{V_M} = \frac{a_M^{eq} + a_M^{sw}}{V_M} = -\frac{M}{V_M}\mathrm{sign}(s) \tag{32}$$

Then the change rate of the leading angle satisfies,

$$\dot{\sigma}_M = \dot{\gamma}_M - \dot{\lambda} = -\frac{M}{V_M}\mathrm{sign}(s) \tag{33}$$

Therefore, when the leading angle $\sigma_M = 0$ but $s \neq 0$, $\dot{\sigma}_M \neq 0$, it means that the leading angle is not an attractor. At the same time, it can be seen from Equation (33) that when the sliding surface $s > 0$, the change rate of the leading angle $\dot{\sigma}_M < 0$, the leading angle decreases; when the sliding surface $s < 0$, the change rate of the leading angle $\dot{\sigma}_M > 0$, the leading angle increases. This means that only when the sliding surface $s = 0$ and the leading angle $\sigma_M = 0$, the leading angle $\sigma_M = 0$ is an attractor of the system. For the leading angle $\sigma_M \neq 0$, the stability of Lyapunov function has been proved by Equation (27).

At the same time, it should be noted that when the leading angle $\sigma_M = 0$, this can be known according to the law of Robida:

$$\lim_{\sigma_M \to 0} \frac{\sin \sigma_M - \sigma_M \cos \sigma_M}{\sin^2 \sigma_M} = \lim_{\sigma_M \to 0} \frac{\sigma_M}{2\cos \sigma_M} = 0 \tag{34}$$

$$\lim_{\sigma_M \to 0} \frac{\sigma_M}{\sin \sigma_M} = \lim_{\sigma_M \to 0} \frac{1}{\cos \sigma_M} = 1 \tag{35}$$

Therefore, the coefficients $K_2$ and $K_3$ are not singular. When the UAV's leading angle is zero, the guidance law can also be activated. According to the above analysis, the guidance law, Equation (24), is a non-singular sliding mode guidance law with impact time control, which is recorded as NSMG.

### 3.2.3. The Extension of the Guidance Law under the Maneuvering Target

For maneuvering target, in order to achieve the effective attack on the target under the designated time, the strategy of predicting the collision point is adopted. The target prediction point $(x_{TP}, y_{TP})$ is shown in Figure 2.

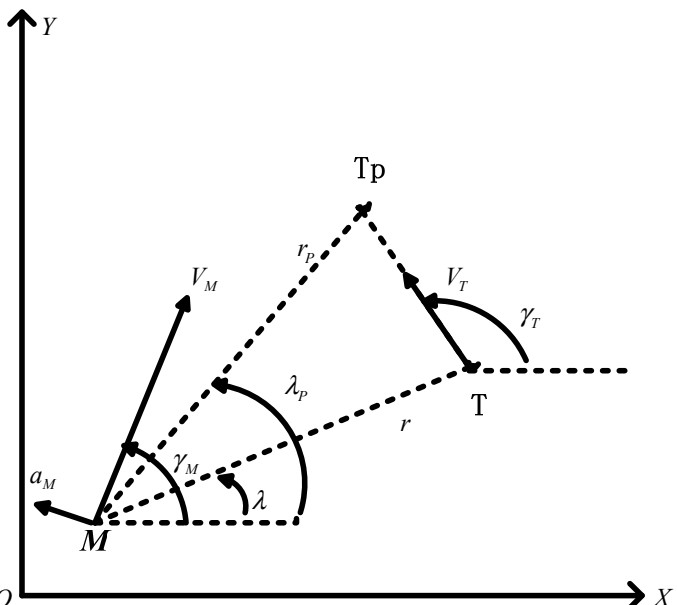

**Figure 2.** Relative motion relationship based on predicted collision point.

The coordinates of the target prediction collision point can be expressed as:

$$
\begin{aligned}
x_{TP} &= x_T + (V_T \cos \gamma_T) t_{go} \\
y_{TP} &= y_T + (V_T \sin \gamma_T) t_{go}
\end{aligned}
\tag{36}
$$

where, $(x_T, y_T)$ is the target coordinates at the current time, $V_T$ is the target velocity, $\gamma_T$ is the target flight path angle. $r_P$ is the relative distance between the UAV and predicted collision point, $\lambda_P$ is the corresponding leading angle. By replacing $r$ and $\lambda$ with $r_P$ and $\lambda_P$ respectively, and bringing them into the guidance law Equation (24), it can attack the maneuvering target in the designated time.

## 4. The Upper-Level Distributed Coordination Strategy Based on Consistency

When the upper layer of the two-layer guidance architecture adopts the centralized coordination strategy, the coordination variable only exists in one member of the formation, which is easier to realize. However, this strategy requires the information of the global system, and when the centralized cooperative member is attacked and fails, the cooperative control of the whole formation will fail, which reduces the robustness and reliability of the system. Therefore, a distributed cooperative guidance law based on consistency theory is designed in this paper.

Let us assume that $n$ UAVs launched simultaneously are required to carry out a saturation attack on a fixed target at the same time. The formation structure composed of multiple UAVs is regarded as the network communication topology structure, and each UAV is regarded as the network communication node. The acceleration command of the *ith* UAV is expressed as:

$$
a_{M,i} = a_{M\_1,i} + a_{M\_2,i} (i = 1, \ldots, n)
\tag{37}
$$

where, $a_{M\_1,i}$ is the local control term of the *ith* UAV for zero miss distance, and $a_{M\_2,i}$ is the cooperative control item for realizing cooperative attack. The local control item selected in this paper is:

$$
a_{M\_1,i} = a_{M,i}^{eq} = -\frac{V_{M,i}}{K_{2,i} + K_{3,i}} \left( (1 + K_{1,i}) + (K_{2,i} + K_{3,i}) \frac{V_{M,i} \sin \sigma_{M,i}}{r_i} \right)
\tag{38}
$$

The collaborative control item can be designed as:

$$a_{M\_2,i} = f\left(s_{i1}(t)t_{go,1}, ..., s_{iq}(t)t_{go,q}, ..., s_{in}(t)t_{go,n}\right) \tag{39}$$

where, $s_{ij}(t)$ is the function of time, and $f$ is the network communication connection. At time $t$, when the *jth* UAV can receive the information transmitted by the *ith* UAV, $s_{ij}(t) = 1$, otherwise, $s_{ij}(t) = 0$, and $s_{ii}(t) = 1$. Then the instantaneous communication matrix describing the information exchange between UAVs in formation can be defined as:

$$S(t) = \begin{bmatrix} s_{11}(t) & s_{12}(t) & ... & s_{1n}(t) \\ s_{21}(t) & s_{22}(t) & ... & s_{2n}(t) \\ ... & ... & ... & ... \\ s_{n1}(t) & s_{n2}(t) & ... & s_{nn}(t) \end{bmatrix} \tag{40}$$

The following formula can be obtained by deriving the time-to-go:

$$\begin{aligned} \dot{t}_{go,i} &= K_{1,i} + (K_{2,i} + K_{3,i})\dot{\sigma}_{M,i} \\ &= K_{1,i} + (K_{2,i} + K_{3,i})\left(\frac{a_{M,i}}{V_{M,i}} + \frac{V_{M,i}\sin\sigma_{M,i}}{r_i}\right) \\ &= K_{1,i} + (K_{2,i} + K_{3,i})\frac{V_{M,i}\sin\sigma_{M,i}}{r_i} + (K_{2,i} + K_{3,i})\frac{a_{M\_1,i}}{V_{M,i}} + (K_{2,i} + K_{3,i})\frac{a_{M\_2,i}}{V_{M,i}} \\ &= f_{1,i}(r_i, V_{M,i}, \sigma_{M,i}) + f_{2,i}(r_i, V_{M,i}, \sigma_{M,i})a_{M\_2,i} \end{aligned} \tag{41}$$

where,$f_{1,i}(r_i, V_{M,i}, \sigma_{M,i}) = K_{1,i} + (K_{2,i} + K_{3,i})\frac{V_{M,i}\sin\sigma_{M,i}}{r_i} + (K_{2,i} + K_{3,i})\frac{a_{M\_1,i}}{V_{M,i}}, f_{2,i}(r_i, V_{M,i}, \sigma_{M,i}) = (K_{2,i} + K_{3,i})/V_{M,i}$.

When $a_{M\_2,i} = 0$, it means that there is no need to adjust the impact time of the UAV, so the impact time of the UAV satisfies $\dot{t}_{f,i} = 0$. From $t_{go,i} = t_{f,i} - t_i$, it can be known that $\dot{t}_{go,i} = -1$. Considering that $f_{1,i}(r_i, V_{M,i}, \sigma_{M,i})$ does not explicitly contain $a_{M\_2,i}$, then $f_{1,i}(r_i, V_{M,i}, \sigma_{M,i}) = -1$. Therefore, whether $a_{M\_2,i}$ is zero or not, the derivative of the time-to-go can be expressed as:

$$\dot{t}_{go,i} = -1 + f_{2,i}(r_i, V_{M,i}, \sigma_{M,i})a_{M\_2,i} \tag{42}$$

The dynamic change of the UAV's impact time can be expressed as:

$$\dot{t}_{f,i} = f_{2,i}(r_i, V_{M,i}, \sigma_{M,i})a_{M\_2,i} \tag{43}$$

For the dynamic system described in Equation (43), according to the cooperative control theory, the following cooperative control algorithm is designed:

$$a_{M\_2,i} = f_{2,i}^{-1}(r_i, V_{M,i}, \sigma_{M,i})\left(\sum_{j=1}^{n}\frac{s_{ij}t_{f,j}}{\sum_{j=1}^{n}s_{ij}} - t_{f,i}\right) = f_{2,i}^{-1}(r_i, V_{M,i}, \sigma_{M,i})\sum_{j=1}^{n}\frac{s_{ij}}{\sum_{j=1}^{n}s_{ij}}\left(t_{f,j} - t_{f,i}\right) \tag{44}$$

By using this algorithm, the operational requirement of impact time cooperative guidance can be satisfied. Substitute Equation (44) into Equation (43) to obtain,

$$\dot{t}_{f,i} = \sum_{j=1}^{n}\frac{s_{ij}}{\sum_{j=1}^{n}s_{ij}}\left(t_{f,j} - t_{f,i}\right) = \sum_{j=1}^{n}d_{ij}\left(t_{f,j} - t_{f,i}\right) \tag{45}$$

where, $d_{ij} = s_{ij}/\sum_{j=1}^{n}s_{ij}$

For the first-order closed-loop cooperative control system described in Equation (45), the research results of the consistent cooperative control theory show that the necessary and sufficient conditions for the communication network topological system to converge

to consistency are as follows: if and only if the communication network topology of the system is strongly connected, that is, there is connectivity between any two nodes in the communication network structure. Therefore, for the cooperative guidance system composed of multiple UAVs, the ultimate goal of the *ith* UAV can be expressed as:

$$t_{f,i} \rightarrow \sum_{j=1}^{n} s_{ij} t_{f,j} / \sum_{j=1}^{n} s_{ij} \tag{46}$$

A non-singular sliding mode guidance law with impact time constraint is designed in Section 3. When the desired impact time is designated in advance, the guidance law can be used to attack the target at the designated time. Based on this, the collaborative control is designed:

$$a_{M\_2,i} = k \varepsilon_i \sin \sigma_{M,i} + M(p \text{sign}(K_{2,i} + K_{3,i}) + 1) \text{sign}(\varepsilon_i) \tag{47}$$

where, $\varepsilon_i = \sum\limits_{j=1}^{n} d_{ij}\left(t_{f,j} - t_{f,i}\right) = \sum\limits_{j=1}^{n} d_{ij}\left(t_{go,j} - t_{go,i}\right)$

To sum up, the distributed time cooperative guidance law with time constraint designed in this paper can be expressed as:

$$\begin{aligned} a_{M,i} &= a_{M\_1,i} + a_{M\_2,i} \\ &= -\frac{V_{M,i}}{K_{2,i}+K_{3,i}}\left((1 + K_{1,i}) + (K_{2,i} + K_{3,i})\frac{V_{M,i}\sin\sigma_{M,i}}{r_i}\right) + \\ &\quad k\varepsilon_i \sin \sigma_{M,i} + M(p\text{sign}(K_{2,i} + K_{3,i}) + 1)\text{sign}(\varepsilon_i) \end{aligned} \tag{48}$$

It can be seen from Equation (48) that the architecture of the distributed time cooperative guidance law is a two-layer cooperative guidance architecture. The bottom layer is the guidance law based on sliding mode control theory, and the upper layer is the distributed cooperative strategy based on consistency.

## 5. Numerical Simulation

### 5.1. Comparison of Methods for Time-to-Go Estimation

The time-to-go estimation methods proposed in this paper and in [32] can be expressed by Equations (17) and (18), respectively. In order to compare the accuracy of the two methods, the navigation gain is set to $N = 3$; the initial range is set to $r_0 = 10,000$ m; the constant velocity of the UAV is set to $V_M = 330$ m/s. The time-to-go calculated by the two methods is compared with the actual time-to-go, which can be shown in Figure 3.

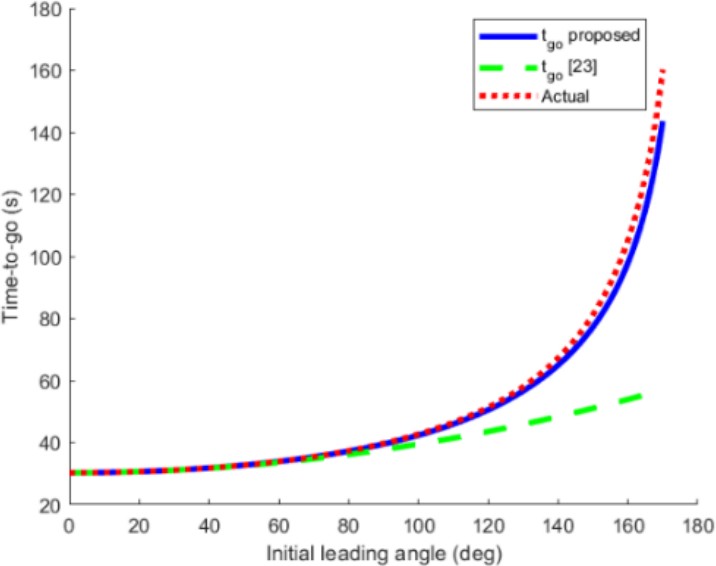

**Figure 3.** Time-to-go with different initial leading angles.

It can be seen from the figure that when the leading angle is less than 60 degrees (deg.), the time-to-go calculated by the two methods is close to the actual time-to-go; however, when the leading angle is greater than 60 deg., the time-to-go calculated by the method proposed in this paper is almost the same as the actual time-to-go, while the time-to-go calculated by the method in [32] is quite different from the actual time-to-go, so the time-to-go estimation method proposed in this paper is more accurate.

*5.2. Verification of the Bottom Non-Singular Sliding Mode Guidance Law*

In order to fully verify the non-singular sliding mode guidance law with impact time control designed in this paper, the following simulation examples are designed for simulation verification.

5.2.1. Comparison of Non-Singular Sliding Mode Guidance Law (NSMG) Law and Sliding Mode Control (SMC) Law

In this case, the performance of the NSMG law and the SMC law are compared. The velocity of the UAV is 330 m/s, the initial position is $(0,0)$ m, the initial flight path angle is 0 deg. The navigation gain is set to $N = 3$. The maximum acceleration of the UAV is 5 g and $g = 9.8$ m/s$^2$. The target position is $(10,0)$ km and the designated impact time is set to 45 s.

$$
\begin{aligned}
a_M &= a_M^{eq} + a_M^{dis} \\
&= \left[ \left\{ 1 + \frac{\dot{r}}{V_M} \left[ 1 + \frac{\sigma_M^2}{2(2N-1)} \right] + \frac{-r\dot{\lambda}\sigma_M}{(2N-1)V_M} \right\} \times C\text{sign}\left( \dot{\lambda} \right) - \frac{2\dot{r}\dot{\lambda}}{r} \right] / \left[ \frac{\cos\sigma_M}{r} - \frac{Cr\sigma_M\text{sign}\left( \dot{\lambda} \right)}{(2N-1)V_M^2} \right] + K_M^{dis}\text{sign}(S)
\end{aligned}
\tag{49}
$$

where

$$
K_M^{dis} = M / \text{sign}\left[ \frac{\cos\sigma_M}{r} - \frac{Cr\sigma_M\text{sign}\left( \dot{\lambda} \right)}{(2N-1)V_M^2} \right]
\tag{50}
$$

$C$ and $M$ are positive constants, and the form of the guidance law is denoted as SMC. The non-singular sliding mode guidance law denoted NSMG proposed in this paper is compared with the guidance law denoted SMC proposed in [33] for simulation, and the variation curves of the UAV's flight trajectory, leading angle, impact time error and acceleration command over time are obtained, as shown in Figure 4.

It can be seen from Figure 4c that the SMC law cannot make the UAV attack the target at the designated time. At this time, the acceleration of the UAV is 0, and the UAV directly flies to the target at a constant speed with a flight time of 30.3 s, which shows that when the leading angle is 0, the SMC law cannot be started, while the NSMG law can attack the target at the designated time. Therefore, when the initial leading angle is 0, the performance of the non-singular sliding mode guidance law with impact time control proposed in this paper is better than the SMC law.

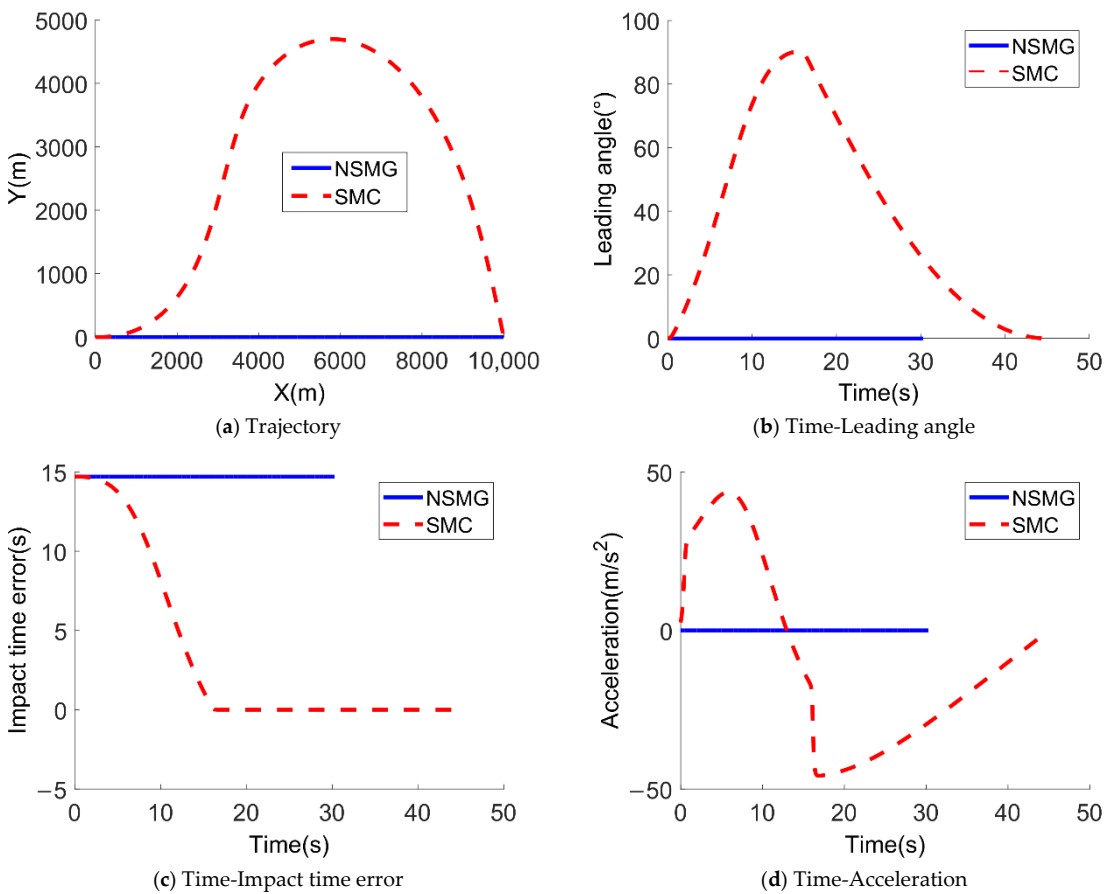

**Figure 4.** Simulation results of the non-singular sliding mode guidance (NSMG) law and the sliding mode control (SMC) law.

### 5.2.2. Performance of the NSMG Law with Different Impact Time

To evaluate the performance of the NSMG law under different impact time, the designated impact time is set to 45 s, 65 s, 85 s and 105 s, respectively. Other parameters are the same as the parameters used in Section 5.2.1. Simulation results are shown in Figure 5.

The legends represent different simulation situations. For example, "$t_d$ = 45 s" represents the simulation results obtained by using the NSMG law when the designated impact time is 45 s.

As can be seen from Figure 5a, the UAV can attack the target at a designated time. The longer the designated impact time is, the more obvious the lateral maneuverability of the UAV will be. As can be seen from Figure 5b,c, in the initial stage, since the estimated value of the UAV's time-to-go is less than the desired time-to-go, the leading angle increases and then gradually converges to zero. Therefore, the corresponding acceleration command increases in the initial stage, and then converges to zero with the decrease of the leading angle, as shown in Figure 5d. At the same time, when the designated time is small, that is, the error of the initial impact time is small, although the acceleration command of the UAV increases in the initial stage, it does not exceed the boundary of the acceleration command. However, when the designated time is large, that is, the error of initial impact time is large, the acceleration command of the UAV will reach the specified boundary in the initial stage, and the larger the error of initial impact time is, the longer the duration will be.

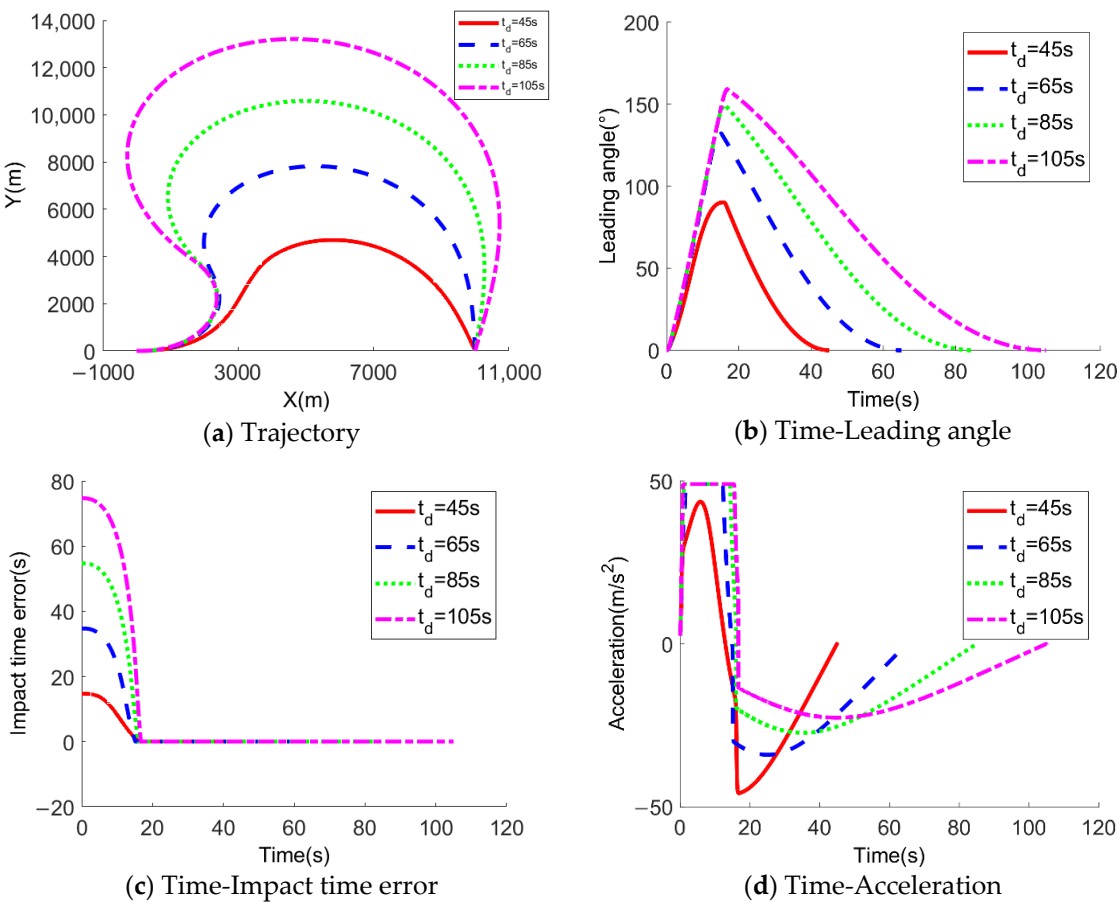

**Figure 5.** Simulation results under different designated impact time.

### 5.2.3. Performance of the NSMG Law with Different Initial Leading Angles

The initial leading angles are set to 20, 40, 60 and 80 deg., respectively. The designated impact time is set to 45 s. Other parameters are the same as the parameters used in Section 5.2.1. Simulation results are shown in Figure 6.

The legends represent different simulation situations. For example, "" represents the simulation results obtained by using the NSMG law when the initial leading angle is 20 deg.

It can be seen from Figure 6a that for different initial leading angles, including the case of large initial leading angle, the UAV can reach the target in a designated time. It can be seen from Figure 6b,c that the leading angle increases in the initial stage to extend the flight time and reduce the impact time error. When the designated impact time is fixed, the larger the initial leading angle is, the smaller the initial impact time error will be, the faster the convergence speed of the impact time error will be, and the smaller the corresponding acceleration command will be. When the impact time error converges to 0, the UAV will fly with pure proportional guidance. When it reaches the target, the relative distance between the UAV and the target is 0, the leading angle is 0, and the acceleration also converges to 0.

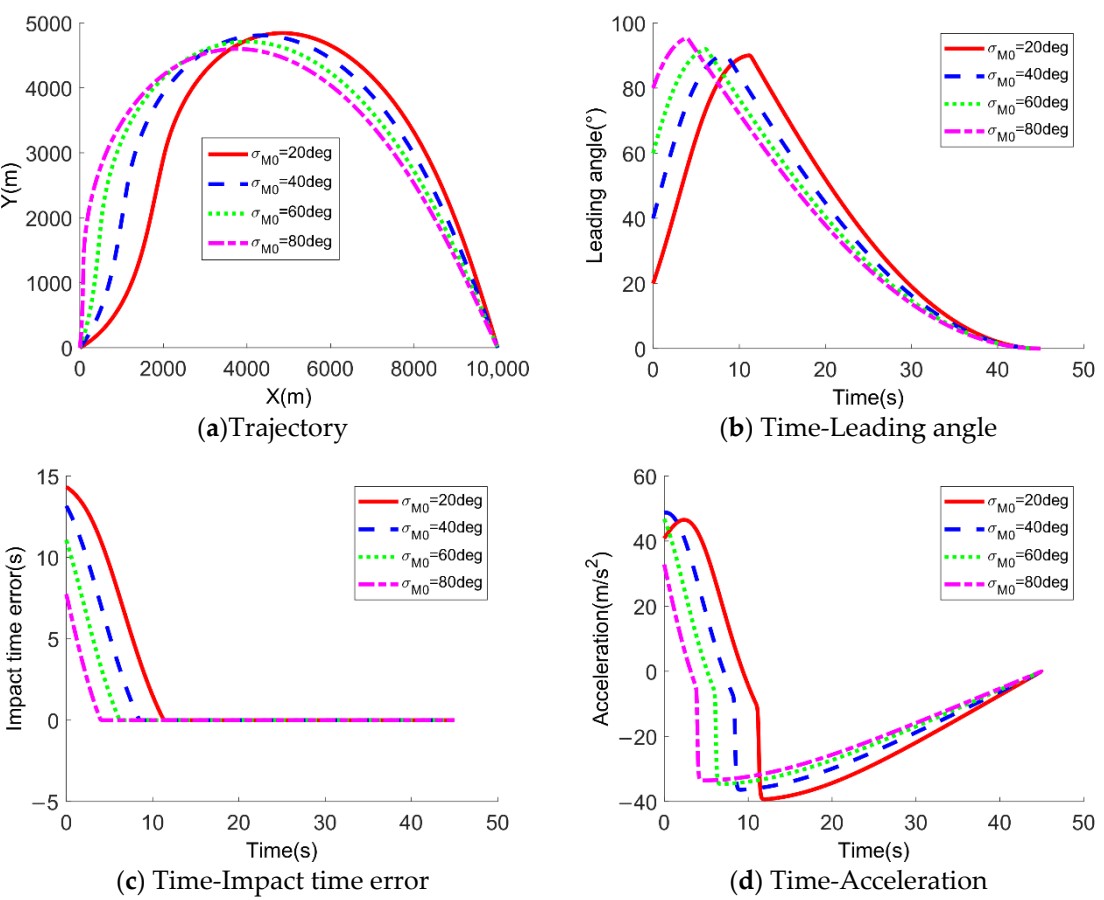

**Figure 6.** Simulation results under different the initial leading angles.

5.2.4. Salvo Attack on Maneuvering Target with the NSMG Law

　　The above simulation examples show that the non-singular sliding mode guidance law NSMG can be applied to strike missions under different initial conditions and different designated impact time. Therefore, the NSMG law can be applied to cooperative combat scenarios. At the same time, in order to verify the effectiveness of the extended guidance law for a maneuvering target, it is assumed that four UAVs with different initial conditions attack the same uniformly moving target. The proportional guidance coefficients are all 3, and the initial launch time is consistent. The other simulation parameters are shown in Table 1.

**Table 1.** Simulation parameters of multiple UAVs' cooperative strike against maneuvering target.

| UAVs/Target | Initial Position (km) | Velocity (m/s) | Initial Flight Path Angle (deg.) | Designated Time (s) |
|:---:|:---:|:---:|:---:|:---:|
| M1 | (0,0) | 330 | 0 | |
| M2 | (5,8) | 320 | 30 | |
| M3 | (15,5) | 310 | 5 | 45 |
| M4 | (5,−8) | 300 | 45 | |
| Target | (10,0) | 50 | 30 | |

　　When all four UAVs adopt the extended form of guidance law under maneuvering target in Section 3.2, the simulation results are shown in Figure 7.

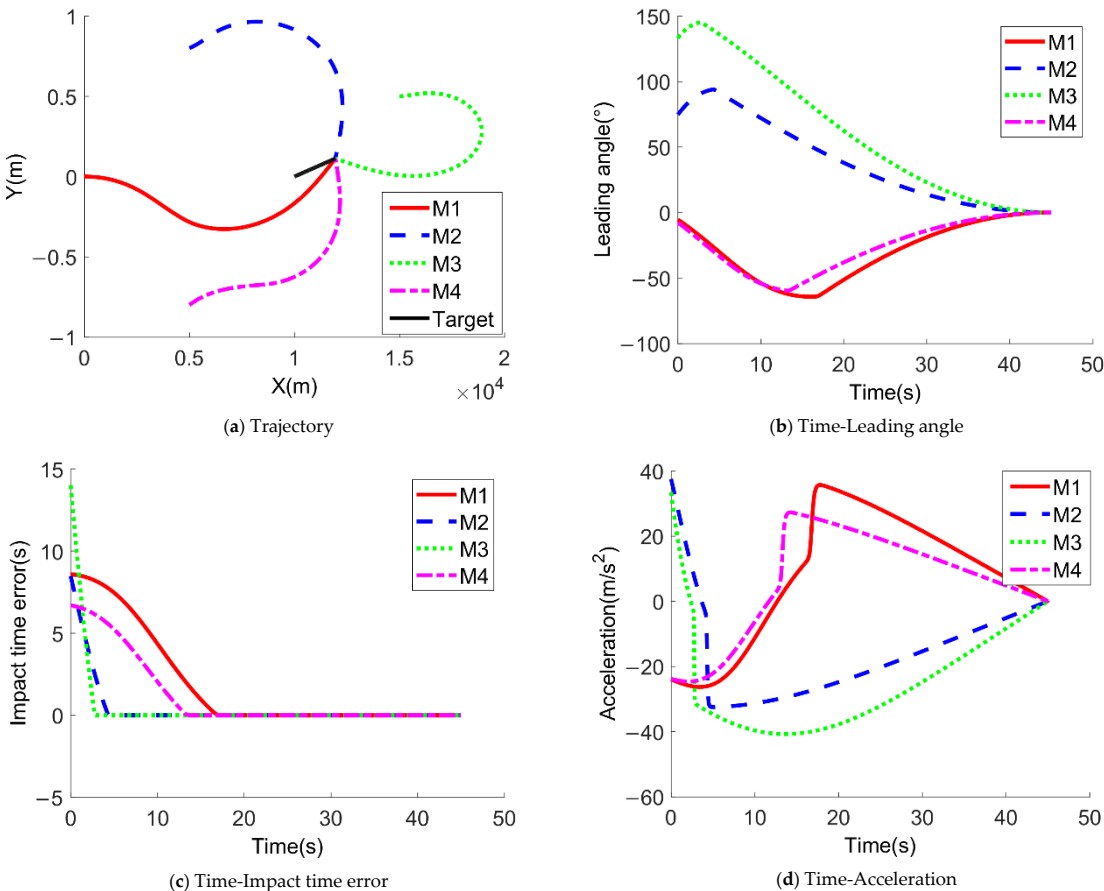

**Figure 7.** The simulation results of cooperative attack on maneuvering targets with NSMG.

Figure 7a–d show the variation curves of the flight trajectory, leading angle, impact time error and acceleration command of four UAVs over time when the NSMG law is applied to a cooperative attack scenario. It can be seen from the figures that the four UAVs under different initial conditions can strike the target at the same designated time, meeting the demand of time coordination. The terminal leading angle of each UAV is 0, and the corresponding acceleration command is also 0. At the initial moment, the estimated value of the time-to-go of each UAV is less than the desired time-to-go. Under the action of the acceleration command, the amplitude of the leading angle increases to extend the flight time, and finally the impact time error gradually converges to 0.

In conclusion, the impact time control cooperative guidance law based on sliding mode control theory has been fully verified, and the guidance law is suitable for strike missions under different initial conditions and different designated impact times. For maneuvering targets, the predictive collision point strategy is used to extend the form of the guidance law, which can realize an accurate attack.

### 5.3. Verification of Upper Level Distributed Coordination Strategy

Let us assume that three UAVs form a network formation to attack the same fixed target, and all UAVs are required to attack the target at the same time. The proportional guidance coefficient of each UAV is 3, and the maximum acceleration is no more than 5 g. The other simulation parameters are shown in Table 2.

**Table 2.** Simulation parameters of upper distributed coordination strategy.

| UAVs | Initial Position (km) | Velocity (m/s) | Initial Flight Path Angle (deg.) | Target Position (km) |
|------|------|------|------|------|
| M1 | (0,0) | 330 | 0 | |
| M2 | (5,8) | 320 | 30 | (10,0) |
| M3 | (15,5) | 310 | −120 | |

Suppose that the network communication matrix of the UAV formation has the following three forms.

$$S_1 = \begin{bmatrix} 1 & 1 & 1 \\ 1 & 1 & 1 \\ 1 & 1 & 1 \end{bmatrix} \qquad S_2 = \begin{bmatrix} 1 & 1 & 0 \\ 1 & 1 & 1 \\ 0 & 1 & 1 \end{bmatrix} \qquad S_3 = \begin{bmatrix} 1 & 1 & 1 \\ 1 & 1 & 0 \\ 1 & 0 & 1 \end{bmatrix} \tag{51}$$

The corresponding network topologies of the three communication matrices are shown in Figure 8.

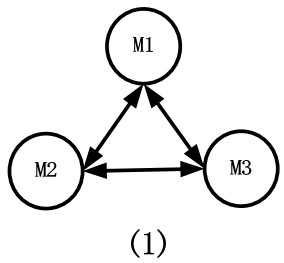

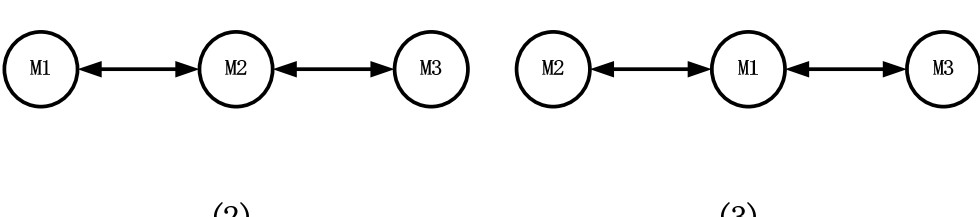

(1)                      (2)                      (3)

**Figure 8.** Different network topologies.

As can be seen from the figure, in the first network topology, the three UAVs can exchange information with each other, which can be called the ring network topology. However, in the second and third network topologies, all interconnections cannot be realized. These two forms can be called chained network topologies. The following is the simulation verification for different network topologies.

### 5.3.1. Ring Network Topology

Based on the double-layer cooperative guidance architecture, when the network topology of the formation is a loop, the variation curves of the flight trajectory, the leading angle, time-to-go and acceleration command of each member of the formation obtained by simulation over time are shown in Figure 9.

As can be seen from Figure 9, the initial time-to-go of the three UAVs is different, respectively, 30.29 s, 37.97 s and 22.96 s. However, after mutual coordination, they gradually become consistent in about 12 s. In the later stage, the cooperative guidance law degenerates into the UAV's own control item, and finally the saturation attack on the target is carried out simultaneously in 33.86 s. The leading angle and acceleration command converge to 0 at the terminal moment.

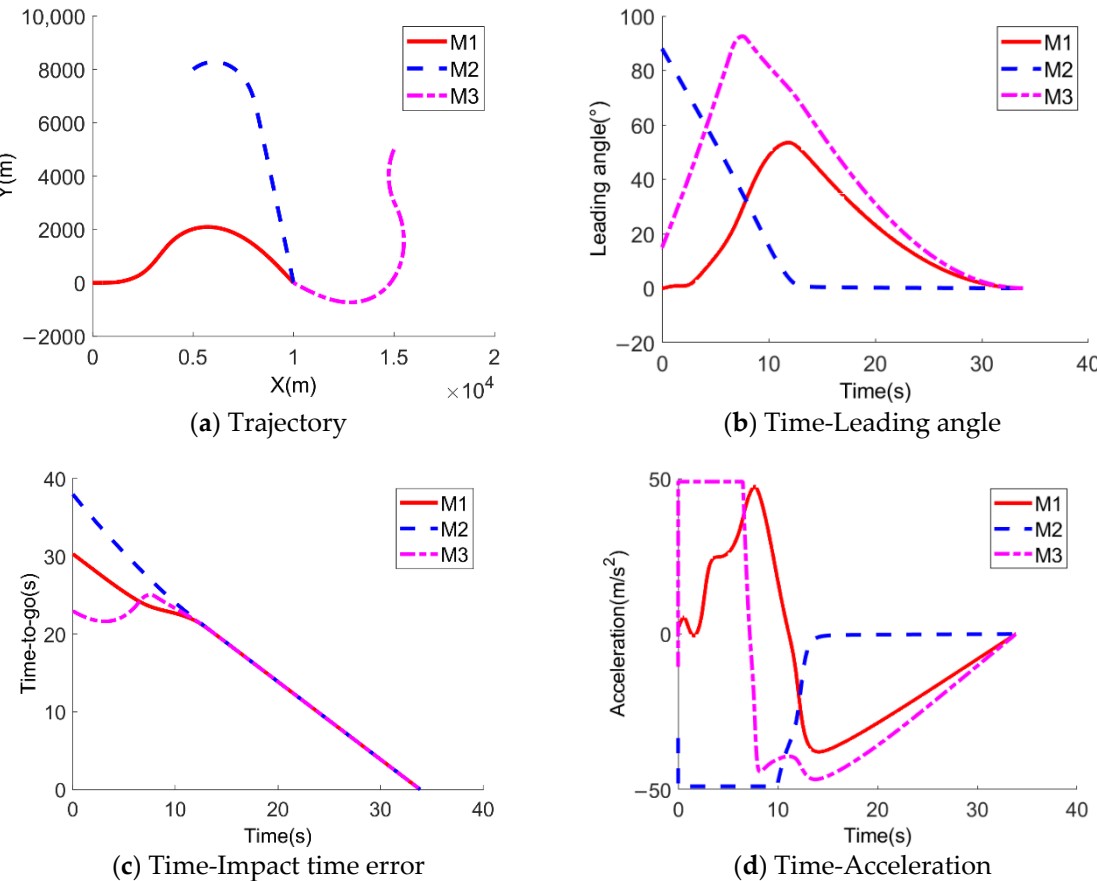

**Figure 9.** Simulation results of cooperative attack under ring network topology.

5.3.2. Chain Network Topology

Taking the chain network topology as an example, the simulation curves of the flight trajectory, the leading angle, the time-to-go and the acceleration command of each member of the formation are shown in Figure 10.

As can be seen from Figure 10, the initial time-to-go of the three UAVs is different, namely 30.29 s, 37.97 s, and 22.96 s. However, after mutual coordination, they gradually become consistent in about 8 s. In the later stage, the cooperative guidance law degenerates into the UAV's own control item, and finally the saturation attack on the target is carried out simultaneously in 34.44 s. The leading angle and acceleration command converge to 0 at the terminal moment.

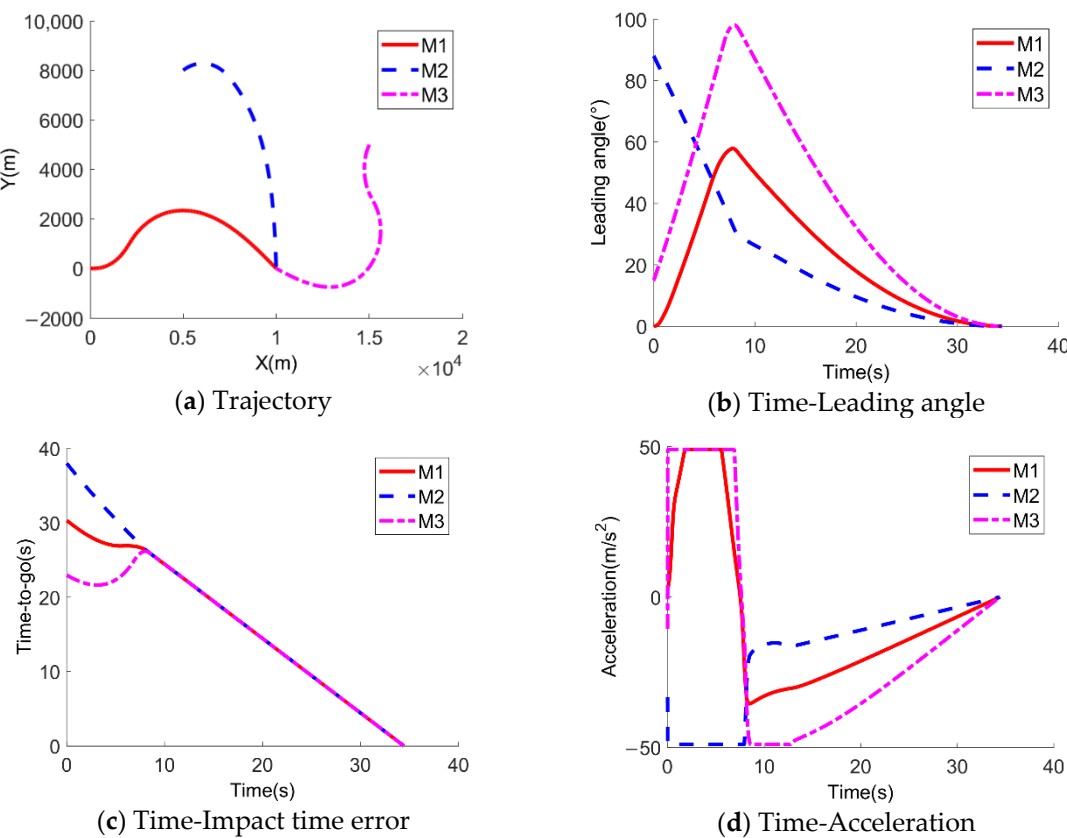

**Figure 10.** Simulation results of cooperative attack under chain network topology.

### 5.3.3. The Situation of Network Topology Switching

In order to verify the time characteristics of cooperative attack in the case of the switching network communication topology of a multiple UAV formation, it is assumed that there is a switching network topology among the above three structures with a switching period of 5 s and a switching sequence of (1) → (2) → (3) → (1). The simulation curves of the flight trajectory, the leading angle, the time-to-go and the acceleration command over time of each member of the formation are shown in Figure 11.

As can be seen from Figure 11, the initial time-to-go of the three UAVs is different at, respectively, 30.29 s, 37.97 s and 22.96 s. However, after mutual coordination, they gradually become consistent in about 11 s. In the later stage, the cooperative guidance law degenerates into the UAV's own control item, and finally the saturation attack on the target is carried out simultaneously in 34.21 s. The leading angle and acceleration command converge to 0 at the terminal moment. It is worth noting that from Figure 11d, it can be seen that the acceleration commands of M1 at 10 s, M2 at 10 s and 15 s have obvious jumps. This is mainly because the coordination information obtained by the UAV has obvious changes when the network topology is switched.

Based on the simulation results of the above three different situations, it can be seen that the upper-layer distributed coordination strategy and the lower-layer non-singular sliding mode guidance law designed in this paper can realize the impact time cooperative guidance under the fixed or switching network topology of the UAV formation through the information exchange between them.

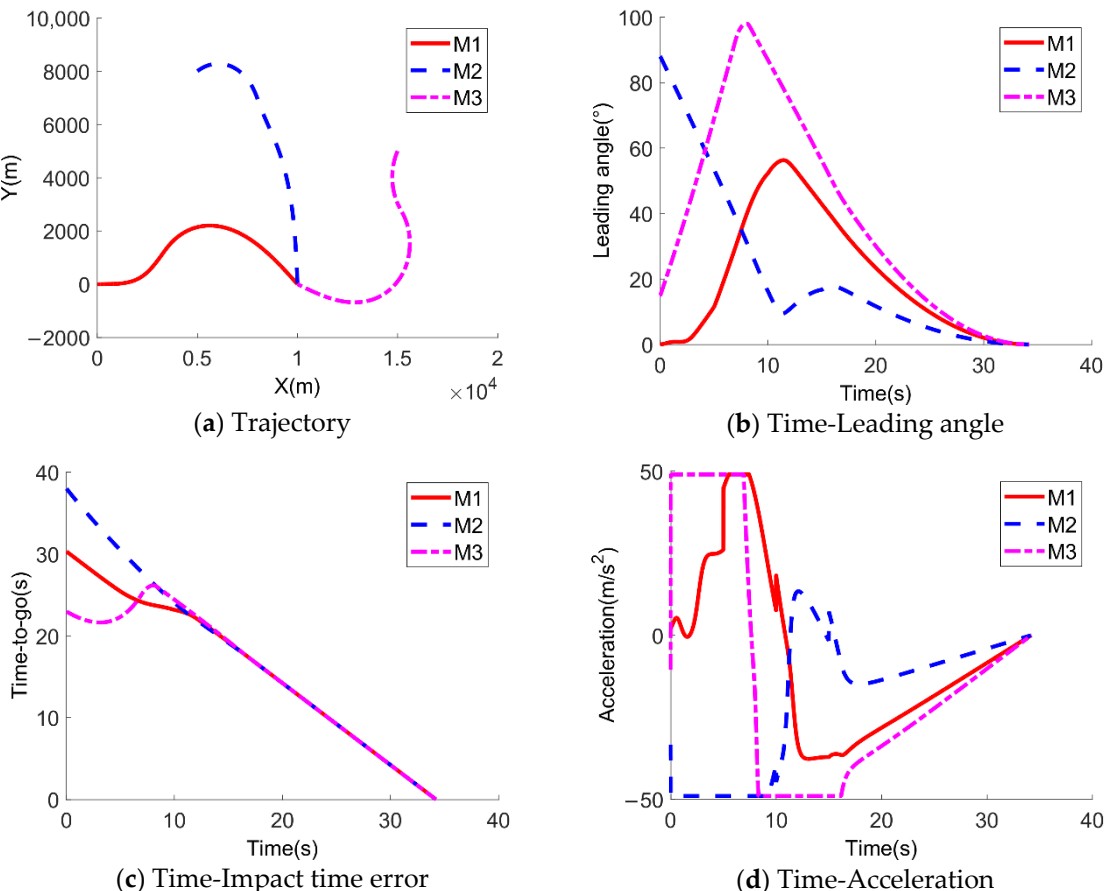

**Figure 11.** Simulation results of switching the network topology.

## 6. Conclusions

In order to solve the problem of system instability when the centralized cooperative strategy is adopted, a distributed cooperative guidance law is designed based on the consistency theory. The structure of the guidance law consists of two parts: the bottom non-singular sliding mode guidance law and the upper distributed coordination strategy. First, a new time-to-go estimation method is proposed, which is more accurate than the existing methods when the leading angle is large. Second, a non-singular sliding mode guidance law (NSMG) of impact time control with equivalent control term and switching control term is designed, which still appears to have excellent performance even if the initial leading angle is zero. Then, based on the predicted crack point strategy, the NSMG law is extended to attack maneuvering targets. Finally, adopting hierarchical cooperative guidance architecture, a terminal distributed cooperative guidance law based on consistency theory is designed. The simulation results show that:

(1) The time-to-go estimation method proposed in this paper is more accurate than [27] under large leading angles.

(2) The non-singular sliding mode guidance law with impact time constraint at the bottom layer can be applied to strike missions under different initial conditions and designated impact time. For maneuvering targets, the predictive collision point strategy is used to extend the form of the guidance law, which can still achieve precise strike.

(3) In this paper, the upper-layer distributed coordination strategy and the lower-layer non-singular sliding mode guidance law are combined to make the formation members exchange information with each other, so as to realize the time cooperative online closed-loop guidance under the condition of fixed or switching network topology of the UAV formation.

**Author Contributions:** Conceptualization, Z.J. and J.G.; Formal analysis, Q.X. and Z.J.; Methodology, Z.J., T.Y. and J.G.; Software, Z.J.; Validation, Z.J. and J.G.; Writing—original draft, Z.J.; Writing-review and editing, T.Y. and Q.X. All authors have read and agreed to the published version of the manuscript.

**Funding:** This research received no external funding.

**Institutional Review Board Statement:** Not applicable.

**Informed Consent Statement:** Not applicable.

**Data Availability Statement:** Not applicable.

**Conflicts of Interest:** The authors declare no conflict of interest.

## Nomenclature

*Acronyms*

| | |
|---|---|
| UAV | Unmanned Aerial Vehicle |
| NSMG | nonsingular sliding mode guidance |
| PNG | Proportional Navigation Guidance |
| NTSMC | nonsingular terminal sliding mode control theory |
| LOS | line of sight |
| SMC | Sliding mode control |

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
