# Peer review of "Terminal Distributed Cooperative Guidance Law for Multiple UAVs Based on Consistency Theory"

_applsci, doi:10.3390/app11188326_

Round 1

Reviewer 1 Report

Overall, the quality of this article is  good. The logical flow is clear. The theoretical derivation is acceptable. 

  The most important and valuable part of this Manuscript is the theoretical model transformed from Stationery  target to maneuvering target. On this issue, the author uses a reasonable theoretical model.    There are two problems that I hope can be improved.   First, the authors made certain assumptions about the parameters and conditions Throughout the entire manuscript, but did not show the reasons for such assumptions and the degree of consistency with the actual situation.    Secondly, the author has made a relatively rigorous formula derivation. But these formulas are difficult for readers or interested research colleagues to reproduce or follow. I think the writing of the formula should be more detailed, and supplemented by the form of images and pictures. This is convenient for interested peers to follow up.   Generally speaking, this is a relatively high-quality theoretical research work, which has a certain guiding significance for the actual experimental work. I hope that in the next step of the author's work, they can compare the measured data with the theoretical simulation, and further correct and improve the accuracy of the theoretical simulation model. 

Author Response

请参阅附件。

Reviewer 2 Report

The article presents an interesting approach to the problem of group control of unmanned aerial vehicles. However, many things in the research are unclear and could be hopefully explained more thoroughly: 1) The mathematical fonts of formulas look very bad and small, please fix this. I guess, that is some LaTex issue.  2) Please add the list of abbreviations to the end of the article, it is very difficult to read the article, looking for the meaning of another abbreviation in the Introduction.  3) I could not quite understand the problem statement of your research. What criteria do you optimize? Are there any constraints on the control (acceleration)? If there are, why is not there any mention of Dubins car dynamics in the article and references?   All in all, I can not recommend this work in its current form for publication in the journal. The issues I have mentioned above must be fixed before reconsidering this research for publication.

Round 2

Reviewer 2 Report

Thank you very much for revising the manuscript. It has become much clearer and understandable in my opinion. Now I believe that it has been sufficiently improved to warrant publication in Applied Sciences.